# Long-Term Fluorescence Behavior of CdSe/ZnS Quantum Dots on Various Planar Chromatographic Stationary Phases

**DOI:** 10.3390/nano12050745

**Published:** 2022-02-23

**Authors:** Paweł K. Zarzycki

**Affiliations:** Research on Small Sailing Yacht—RoSSY, Dworcowa 11/15, 75-201 Koszalin, Poland; pkzarz@wp.pl

**Keywords:** quantum dots, fluorescence, planar chromatographic stationary phases, detection, time domain analysis, hybrid nanomaterials, Micro-TLC, chemometrics, multivariate statistics

## Abstract

Nanoparticles, particularly quantum dots (QDs), are commonly used for the sensitive detection of various objects. A number of target molecules may be determined using QDs sensing systems. Depending on their chemical nature, physicochemical properties, and spatial arrangement, QDs can selectively interact with given molecules of interest. This can be performed in complex systems, including microorganisms or tissues. Efficient fluorescence enables low exposure of QDs and high sensitivity for detection. One disadvantage of quantum dots fluorophores is fluorescence decay. However, for given applications, this property may be an advantage, e.g., for highly sensitive detection based on correlation images in the time domain. This experimental work deals with the measurement of fluorescence decay of Lumidot TMCdSe/ZnS (530 nm) quantum dots. These nanoparticles were transferred to the surface of various planar chromatographic stationary phases. Fluorescence of formed spots was recorded at room temperature over a long period of time, namely 15.7824 × 10^5^ min (three years). The resulting signal profiles in the time domain were analyzed using classical approach (luminescence model comparison involving different mathematical models).Moreover, fluorescence behavior on different TLC/HPTLC supports was investigated using multivariate statistics (principal component analysis, PCA). Eight planar chromatographic stationary phases were investigated, including cellulose, octadecylsilane, polyamide, silica gel and aluminium oxide in different forms (TLC and HPTLC types). The presented research revealed significantly different and non-linear long-term QDs behavior on these solids. Two different fluorescence signal trajectories were recorded, including typical signal decay after QDs application to the plates and long-term intensity increase. This was particularly visible for given planar chromatographic adsorbents, e.g., cellulose or octadecylsilane. To the author’s knowledge, these findings were not reported before using the stationary chromatographic phases, and enable the design of future experiments toward sensing of low molecular mass chemicals using, e.g., advanced quantification approaches. This may include signal processing computations based on correlation images in the time domain. Additionally, the reported preliminary data indicates that the investigated nanoparticles can be applied as efficient and selective fluorophores. This was demonstrated on micro-TLC plates where separated bioactive organic substances quenching from cyanobacteria extracts were sensitively detected. The described detection protocol can be directly applied for different planar chromatographic systems, including paper-based microfluidic devices, planar electrophoresis and/or miniaturized microfluidic chip devices.

## 1. Introduction

There is increasing interest in developing simple analytical devices, which can generate qualitative or quantitative data that may be acquired and transmitted by digital cameras, especially those built into mobile phones/smartphones devices [1]. Such inexpensive and robust hybrid analytical systems can be easily used in remote sampling areas, particularly for fast screening of various micropollutants and/or biomarkers. Different forms of non-forced flow rate planar chromatography including thin-layer chromatography (TLC), high-performance thin-layer chromatography (HPTLC), or micro-thin-layer chromatography (micro-TLC or micro-HPTLC) working in different modes (e.g., normal or reversed-phase mobile phases, saturated/unsaturated chambers, circular or linear one/two-dimensional development systems) are particularly of interest because TLC analysis results with picture type object [2]. Developed planar chromatograms may be easily photographed, and this form of data can be simply transmitted via internet connections to given research centers. Moreover, planar chromatography can be easily miniaturized and hyphenated with advanced and sensitive detectors, including mass spectrometers, FTIR spectroscopy or electrochemical sensing devices [3,4]. It should be noted that planar chromatography is fairly efficient as a separation tool with several advances on their counterpart: column liquid chromatography (high-performance liquid chromatography; HPLC). They include parallel/multiple samples development, low consumption of chemicals for mobile phases, ability to work without external pumps, degassers, pre-columns, injectors or different large and complex auxiliary devices (mobile phase movement is driven by capillary forces), the possibility for direct analysis of heavily loaded matrices without sample pre-purification (due to single-use of stationary phase), fast-developing mode (in case of micro-TLC, the analysis time can be completed within 1–2 min) [5]. Planar chromatographic analysis can be manual or fully automated, significantly decreasing the relative low repeatability of such analysis [6].

The main challenge for TLC or other planar analytical devices like μPADs (micro-paper analytical devices) quantification systems is sensitive and selective detection. There are several modern mass spectrometry detectors manufactured for use with planar chromatography, for example, DART-MS, direct analysis in real-time mass spectrometry, TLC–MALDI-MS, matrix-assisted laser desorption/ionization mass spectrometry or TLC–EI-MS, electron impact ionization mass spectrometry. These systems may work in atmospheric pressure but are complex, expensive, and usually non-portable. It should be noted that the sensitivity of planar chromatographic systems may be significantly increased by simple post-run derivatization of target analytes and application of zero-background detection methods, e.g., based on fluorescence [7]. Additionally, modern pharmacy and chemistry still commonly use derivatization protocols dedicated to planar chromatography. This is mainly applied for tight control, purity confirmation, and substance identification of pharmaceutical formulations that protocols are described in the General Monographs of the European Pharmacopoeia [6].

Quantum dots (QDs) and QD-derived nanomaterials are particularly of interest in terms of photodetection. These nano-objects have somewhat unique optical properties. They can adsorb multi-wavelength light but then may emit only a narrow peak of a given wavelength. There is a number of quantum dot types, physicochemical structures, and spatial arrangements involving cadmium, silver, gold, or carbon components [8]. They may work in various fluorescence modes, including fluorescence resonance energy transfer (FRET), a powerful tool for biosensing and bioimaging [9]. QD-based sensors can be easily miniaturized and therefore, they may be easily combined with various portable analytical devices [1]. For these reasons, there is an enormous and long-term interest in research and applications of different QDs in biomedicine and analytical chemistry [8,9,10].

In the literature, quantum dots are usually described as photostable objects with good fluorescence stability, and this is usually considered an advantage for many sensing applications [8,9]. However, signal decay is expected for given detection techniques, particularly involving correlation images in the time domain [11]. Such detection setup can enable advanced signal processing, including FT (Fourier transform) calculations and then the computation of contrast images in both time and frequency domains, including phase and amplitude contrast images. Using such a database, the unique correlation images in the time domain may be created and detection sensitivity for given objects can be significantly increased, typically two orders of magnitude in comparison to the detection limit calculated from raw and unprocessed pictures.

The aim of this experimental work is to describe the long-term fluorescence behavior (fluorescence decay) of selected quantum dot nanoparticles transferred to various planar chromatographic supports (stationary phases). This enables the selection of potential detection modes and an analytical setup for future quantification studies using quantum dot-driven fluorescence as a data source for correlation images in a time domain. The presented data and analytical approach enable the fast selection of given separation support for sensitive analytes on planar separation systems, including micro-TLC and/or μ\PAD devices.

## 2. Materials and Methods

### 2.1. Chemicals and Materials

Quantum dot nanoparticles solution Lumidot^TM^ CdSe/ZnS (core-shell type, 530 nm, 5 mg/mL in toluene) was obtained from Sigma-Aldrich Co. (3050 Spruce Street, St. Louis, MO 63103 USA). According to the manufacturer data, the active matrix groups of this product were stabilized with hexadecylamine (HDA) ligand coating surface treatment (capping agent). The absorption max = 530 nm and emission wavelength (peak) = 525–535 nm. Toluene (99.5% A.C.S. Reagent) and acetone (99.9% HPLC grade) were purchased from Sigma-Aldrich (Steinheim, Germany). *n*-Hexane (95%) and methanol (LiChrosolv 99.8% for LC) were products of Fluka Chemie AG (Buchs, Switzerland) and Merck (Darmstadt, Germany), respectively.

All planar chromatographic plates were obtained from Merck (Darmstadt, Germany), cut to size 10 mm × 50 mm, and placed on black support forming an85-mm wide plates array. After applying QDs solutions to these stationary phases, the testing array was stored in a light protected box at room temperature during the experiment performed. Particularly, following planar chromatographic stationary phases were studied: (*i*) Cellulose (glass-based; TLC pre-coated plates; layer thickness 0.1 mm), (*ii*) RP-18 W (glass-based, HPTLC pre-coated plates; layer thickness 0.2 mm), (*iii*) RP-18 W F_254_S (glass-based HPTLC plates; layer thickness 0.2 mm), (*iv*) RP-18 F_254_ S (glass-based HPTLC plates; layer thickness 0.2 mm), (*v*) Polyamide 11 F_254_ (glass-based TLC pre-coated plates; layer thickness 0.15 mm), (*vi*) Silica gel 60 F_254_ (glass-based TLC plates; layer thickness 0.25 mm), (*vii*) Silica gel 60 W F_254_ S (glass-based TLC plates; layer thickness 0.25 mm), (*viii*) Aluminium oxide 60 F_254_ Type E (glass-based TLC plates; layer thickness 0.25 mm).

Spirulina (450 mg capsules containing *Spirulina maxima* desiccated cyanobacteria) was obtained from A-Z Medica (Gdańsk, Poland).

### 2.2. Micro-TLC Plates Application Protocols

For the quantum dots stability study, the TLC plates array (composed of eight different stationary phases cut to size 10 × 50 mm^2^) was spotted with 1 μL of Lumidot solution at a concentration of 5, 0.5, 0.05 and 0.005 mg/mL using a Hamilton syringe. The manual application of 32 spots (8 plates × 4 QDs concentrations) was finished within 12 min, approximately. TLC plates with quantum dots were stored at room temperature in a glass container covered with aluminium foil to protect from light exposure.

For the detection experiment, both quantum dots and spirulina extract lanes were generated by a Linomat 5 semi-automatic application instrument (Camag, Switzerland), controlled through the Planar Chromatography Manager (winCATS software, 1999–2008, version 1.4.4.6337). Using the spray-on technique, a narrow 43-mm-long vertical detection band of quantum dots solution (10 μL, 5 mg/mL) was formed across the start line, which was positioned 7 mm from the bottom edge of a 50-mm-long TLC micro-plate. After that, a 10-mm-long sample band (6 μL of spirulina extract) was formed along the start line.

### 2.3. Micro-TLC Chromatography

A separation and detection experiment involving spirulina extract was performed on a glass-based HPTLC RP18W plate. Before the sample application, the factory-prepared plates (100 × 100 mm^2^) were cut to a working size of 25 × 50 mm^2^. The sample starting line was placed 7 mm from the plate bottom edge, allowing a maximum eluent front migration distance of 43 mm. The detailed protocol for the application of detection and sample lanes was reported above. Micro-planar separation was performed using a homemade temperature-controlled removable horizontal micro-TLC chamber unit as described previously [12]. The system provided a constant micro-TLC plate temperature, set at 30 °C temperature with an accuracy of ±0.02 °C.

Chromatographic separation was performed under unsaturated chamber conditions. The following chamber working protocol was applied to obtain chromatograms: a micro-TLC plate sprayed with quantum dots and samples lanes was positioned horizontally inside a chamber module with the stationary phase layer placed upside down. Afterward, the chamber module was transferred into a temperature-controlled oven cavity and sealed using a 1-mm-thin glass cover. Then, the movable cover of the oven was slid to reach the position above the TLC chamber module, and the temperature equilibration step was performed for 10 min. The chromatographic process was started after injecting of eluent composed of acetone: *n*-hexane, (3:7, *v*/*v*) through an injection pipe into a mobile-phase application bar. Finally, the TLC plate was removed from the chamber module immediately after the mobile-phase front reached the plate edge opposite the application bar. The plate was immediately photographed under different UV-Vis light conditions after stationary phase drying at room temperature for a few minutes.

### 2.4. Spirulina Extraction

Spirulina desiccated material was powdered manually using a small ceramic mortar, and 150 mg samples were transferred into 5 mL glass tubes. The samples were mixed with 1 mL of methanol. Afterward, the tubes were sealed and sonicated for 1 h at room temperature using an ultrasonic bath Sonic 1 (80 W, Polsonic, Warsaw, Poland). Next, the tubes were centrifuged (5 min, 5800 rpm; MPW-53; MPW Med Instruments Spółdzielnia Pracy, Warsaw, Poland) and clear extract was sprayed onto the TLC plate start line.

### 2.5. Data Acquisition

The array of planar chromatographic plates was photographed using an OLYMPUS digital camera CAMEDIA C-5050 ZOOM (Olympus Optical Co., Ltd., Tokyo, Japan) equipped with UV filter (43 mm; Marumi, Tokyo, Japan). The following manual settings were applied: aperture F 8.0; shooting range: normal mode (max zoom, manual focus); ISO 64; white balance: auto; shutter speed 2 s (1/125 s for visible light using internal flash; power = 5); file format: RAW (image resolution 2560 × 1920). The photographed object was placed inside a black box (5 cm × 20 cm × 20 cm) that was illuminated with UV (254/366 nm) Lampa UV-standard (Cobrabid, Warsaw, Poland) consisting of UV sources: UV254 nm PHILIPS TUV 6 W, G6 T5 and UV366 nm PHILIPS TL 6 W/108, BLB F6 T5. The partial arrangement of camera lens-object- lamp was as follow: lens-object distance = 265 mm, lamp-object distance = 230 mm; lens-object-lamp angle = 15 degrees.

The intensity of TLC spots for quantum dots stability study was recorded within 3 years period of time. Image acquisition process was starting 1 min after application of last QDs spot on TLC plates array as well as 60 and 120 min and then after 4, 6, 18, 24.217, 48, 72, 96, 120, 144, 168, 336, 504, 672, 840, 1008, 1200, 1368, 1512, 2184, 2856, 3360, 4200, 4872, 6216, 7728, 8760 (first year), 10,584, 12,768, 16,296, 17,544 h (second year) 21,000, 22,680, 25,032, 26,304 h (third year).

For the detection of spirulina extract components, picture acquisition on the TLC plate was performed immediately when the separation protocol was completed. This was done approximately 30 min after spraying of quantum dots detection lane, which was the first step of the analytical protocol described above.

### 2.6. Data Processing and Analysis

Raw digital pictures were converted to TIFF 8 bit RGB format from which the TIFF 8 bit grayscale pictures were extracted. From these digital files, all of the cross-section and pixel intensity parameters were derived using Scion Image freeware (Scion Corp., Frederick, MD, USA; Version 4.0.3.2; http://www.scioncorp.com/, 27 December 2006) and ImageJ freeware (ver. 1.42q Wayne Rasband, National Institutes of Health, Bethesda, MD, USA; http://rsb.info.nih.gov/ij, 27 February 2017). In the case of QDs, the images and chromatograms included within figures presented in this paper, a global auto-balance conversion filter was applied to enhance the bands/spots contrast for satisfactory computer screen displaying and printing reproduction.

Nonlinear curve fitting concerning intensity in the time domain data was performed using Origin v8.5.1 Win Pro software provided by OriginLab Corporation (Northampton, MA, USA). Quantitative data were inspected with the principal component procedure (PCA) using XLSTAT XLSTAT-Pro/3DPlot statistical and visualization package (version 2008.2.01) obtained from Addinsoft (Paris, France) in conjunction with Microsoft Excel 2002.

## 3. Results and Discussion

Quantum dots based on CdSe or CdTe particles have been found to be efficient sensing components of paper-based analytical devices for detecting toxic inorganic elements, including Cu (II) and complex organic biomolecules like enzymes [13,14]. This is mainly because they are efficient as fluorophores. These nanoparticles can be easily prepared, are relatively non-expensive, and are chemically stable/resistant and therefore, they are still commonly applied for photoluminescent probing and quantitative applications [1,15,16,17,18].

Figure 1A consists of a series of toluene CdSe/ZnS nanoparticles solutions, which were photographed at visible light conditions under illumination by UV light (254/366 nm). As can be seen, an intense green fluorescence can be observed for 50–500 μg/mL solutions; even they are placed inside the ordinary glass vials that are not fully transparent for UV radiation. Figure 1B represents an array of various planar chromatographic lanes in which quantum dots were transferred at different masses, ranging from 0.005 to 5 μg per spot. Under such conditions, the fluorescence response is not disturbed by solvent because all volume of toluene was evaporated after the application of QDs solution to the plates’ surface. For these objects, an intense fluorescence can be registered for QDs mass equal to 5 μg per spot and 366 nm UV illumination. The array exposure for UV 254 nm light results with strong plate background fluorescence, particularly those modified with fluorescence additive (stationary phases No. 3–8). In this case, the QDs are visible as dark spots where fluorescence quenching is dominating. For that reason, subsequent studies should focus on zero-background detection mode using a 366 nm excitation wavelength.

The efficient detection of target components is strongly dependent on the possible contrast range provided by a given sensing system (describing available noise to signal detection space). The proposed detection method involves a 366-nm excitation wavelength, and this system in basic form may work as a simple quenching mode system. In such a case QDs signal will be decreased by target molecules that are concurrently separated on a given TLC plate modified with QDs. To quantify the mass-signal response of QDs spots, which were formed on planar chromatographic plates, appropriate densitometric profiles were generated. As an example, aluminium oxide support was investigated in which regular QDs spots were formed (Figure 2). For this stationary phase-detection limit, the value of QDs (LOD; limit of detection) can be estimated as 0.05 μg per spot. Accordingly, for the densitometric profile for this spot, a maximum signal strength exceeds the background noise level approximately threefold. It is clear to see that if 5 μg of the QDs mass is transferred to the plate surface, the signal strength exceeds 200 (measured as grayscale), in comparison to the 256 maximum value available for a digital camera’s CCD sensor (2^8^ bit RGB recorded pictures). 

From a practical point of view, the densitometric profiles of the pictures and spots presented in Figure 3A strongly indicate that applying a relatively low amount of quantum dots enables forming an efficient photoluminescent zone for the detection of target components on various planar chromatographic plates. Nevertheless, the investigated Lumidot^TM^ CdSe/ZnS quantum dots did not affect the quenching mode in the case of 254 nm excitation wavelength (Figure 3B).

The main question raised in this study was to recognize the real stability of quantum dots fluorescence signal on the planar chromatographic stationary phases over time. Therefore, the planar chromatographic plates array was photographed over a long period of time. The results of this experiment are presented in form of graphs in Figure 4. The top graph represents the recorded profiles of the signal intensity for all planar chromatographic stationary phases within three years. The bottom graph covers the initial recording time within the first 120 h (5 days). As can be seen, the signal from QDs is unstable and strongly non-linear. Particularly, a massive signal deterioration is observed within the first few hours of the experiment, for which the exponential decay pattern can be considered (this is presented in Figure 5). Then, depending on the stationary phase composition, the signal starts to increase. This phenomenon begins after two days for QDs that were transferred to cellulose plate and after 28 days for RP-18 W HPTLC plate. Within the last two years, the fluorescence signal is going to stabilize or decrease slightly. 

According to the data presented in Figure 5, there is a significant difference between the calculated mean lifetime of QDs fluorescence signal. This parameter was computed using data from the first 120 h of the experiment. The corresponding values of the half-life (***T*_1/2_** = *ln* (2) * mean lifetime) are following: 159, 52, 101, 153, 87, 103, 109 and 48 min for plates from 1 to 8, respectively. Based on these data, the most stable system (in terms of signal decay) that can be used for classical fluorescence detection seems to be cellulose and RP-18 F_254_ S HPTLC plates (mean lifetime = 230 and 221 min, respectively). However, for detection involving the generation of correlation images in the time domain that requires signal variation in the time domain, the plates RP 18 W HPTLC and aluminium oxide 60 F_254_ Type E TLC should be preferred, considering the fast changes in signal level within 1 h.

To compare the QDs’ fluorescence within the whole period of time investigated, the multivariate statistics calculations (principal component analysis) were performed. The application of such data mining enables the clustering of similar objects (TLC plates consisting of different stationary phases) based on complex data provided (variables) that may be affected by several multifactor and orthogonal parameters, including: QDs oxidation, reaction with surface ligands or effect of adsorbents spatial structure. 

As a first approach, the initial data set was composed of signal values from QDs on all chromatographic plates and fluorescence measured from a reference area—the paper patches in which the plates labels were printed (they are visible in Figure 1B at the bottom of the each TLC plate strip). These patches were characterized by intense white/blue fluorescence and were photographed simultaneously with the plates array each time during experiment. This reference object was labeled as a black square (No 9) in the graph within Figure 6A. The presented factor scores plot has revealed that taking into account the most important F1 parameter (counting for more than 86% of total variability) the fluorescence behavior of the reference area and remaining objects is different since this sample is away from the QDs samples. Within the QDs’ samples, two clusters are formed: plates 1, 4, 5, 8 and 2, 3, 6, 7. The graph in Figure 6Bconsists of PCA analysis based on re-calculated data without the reference signal area. Objects grouping in this graph strongly suggest that fluorescence profiles over time are significantly different for cellulose (No 1), then for plates No 4, 5, 8, and for No 2, 3, 6, 7. Such plate clustering is visible, taking into accountthe most important factor, F1, which, in this case, explains over 68% of the variability. Interestingly, clusters 2,3,6,7 consist of pure silica gel plates and low-density octadecylsilane plates (C18; RP-18, wettable with water). Within clusters 4,5,8, both high-density C18 plates are included. This strongly suggests that plates polarity may influence the QDs fluorescence behavior on the planar stationary phases over a long period of time.

From a practical point of view, it is interesting how the detection system described above involving QDs fluorescence may work in a complex organic matrix to detect given analytes. For that reason, the spatial arrangement of QDs was proposed, as is visible in Figure 7. As the first step, a narrow quantum dots detection lane (1 mm wide, approximately) was sprayed in parallel to the future mobile phase development on the 50-mm-long micro-TLC plate. As can be seen from the bottom picture presented in Figure 7, the detection line results with strong fluorescent signal and low background signal noise after 366 UV light exposure. On these plates, spirulina extract was transferred on start line and chromatogram was developed using a simple binary mobile phase composed of acetone: *n*-hexane (3:7, *v*/*v*) within temperature-controlled micro-chamber described previously [12]. After the drying of the mobile phase, the spots pattern was photographed at different UV-Vis light conditions. The separation and detection results are presented in Figure 8, where the images of the original micro-TLC plate and the densitometric scans were presented. The spirulina extract contains a number of low-molecular mass organic compounds, mainly carotenes, flavonoids and chlorophyll-related dyes that were well-separated and detected in both visible and UV 254 light (Figure 8 lane 2 densitograms). Under a UV 366-nm light, mainly red fluorescence from chlorophylls and some blue fluorescence from flavonoids can be recorded. It is noteworthy that the QDs detection line affected the TLC separation efficiency, which was slightly deteriorated. However, the observed fluorescence quenching of QDs signal at 366 nm resulted in a massive increase of detection for given components, in comparison to both Vis and UV 254 nm detection mode (lane 1 densitograms). This clearly indicates that the detection sensitivity of the micro-TLC separation system can be increased in the presence of quantum dot nanoparticles.

## 4. Conclusions

The results of presented experimental work clearly indicate that CdSe/ZnS quantum dots can be fairly unstable after contacting the planar chromatographic stationary phases. The fluorescence signal pattern is strongly non-linearover a long period of time. Within the first several hours, exponential decay may be observed, and the calculated values of the mean lifetime are significantly different for the investigated stationary phases. Multivariate calculations revealed that the fluorescence pattern can be similar within three groups of stationary phases: (*i*) cellulose, (*ii*) silica, and low C18 covered TLC plates as well as (*iii*) Aluminium oxide and high-density C18 plates. This strongly suggests that the support polarity may be a key factor for the fluorescence behavior of the adsorbed QDs; however, this hypothesis should be supported by additional experimental data. QDs can form an efficient sensing system for low-molecular-mass organic compounds, mainly chlorophyll dyes, despite the fluorescence signal deterioration. On the other hand, the phenomenon of QD fluorescence decay can be an advantage in case of a detection mode based on the correlation images in the time domain, and this aspect of the research will be investigated in the future.

## Figures and Tables

**Figure 1 nanomaterials-12-00745-f001:**
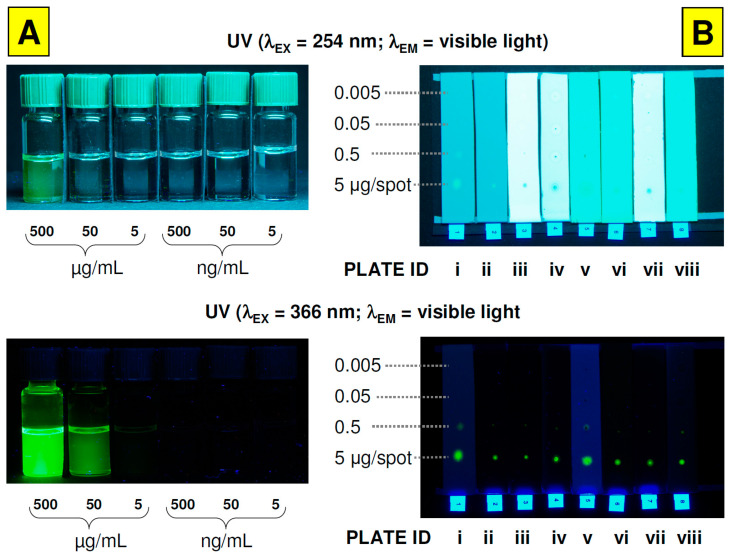
General view of Lumidot^TM^ CdSe/ZnS quantum dots solutions ((**A**); left column) and QDs spots on TLC plates ((**B**); right column) under different UV light conditions. Chromatographic stationary phases (plates) IDs: (*i*) Cellulose TLC, (*ii*) RP-18 W HPTLC, (*iii*) RP-18 W F_254_S HPTLC, (*iv*) RP-18 F_254_ S HPTLC, (*v*) Polyamide 11 F_254_ TLC, (*vi*) Silica gel 60 F_254_ TLC, (*vii*) Silica gel 60 W F_254_ S TLC, (*viii*) Aluminium oxide 60 F_254_ Type E TLC. Digital Camera settings for QDs vials: F = 8.0, shutter speed = 5 s, ISO 64.

**Figure 2 nanomaterials-12-00745-f002:**
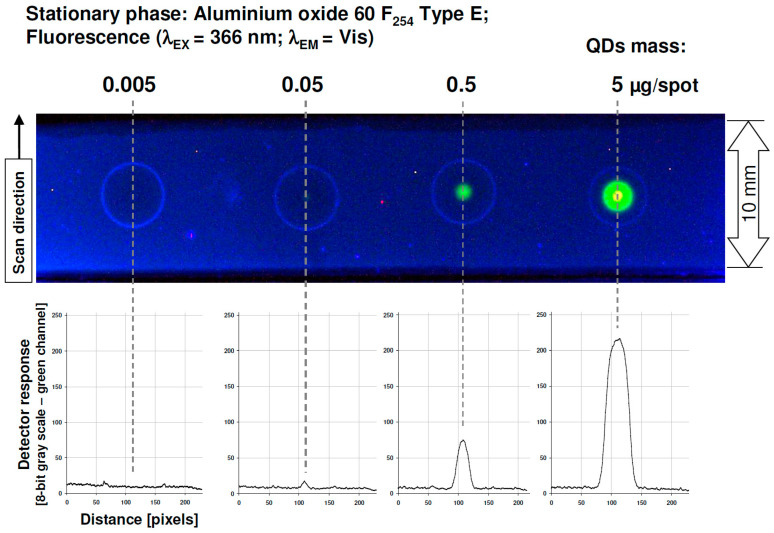
Signal response for different QDs mass transferred to Aluminium oxide 60 F_254_ Type E TLC plate. Intensity profiles (**bottom**) were recorded for the green channel.

**Figure 3 nanomaterials-12-00745-f003:**
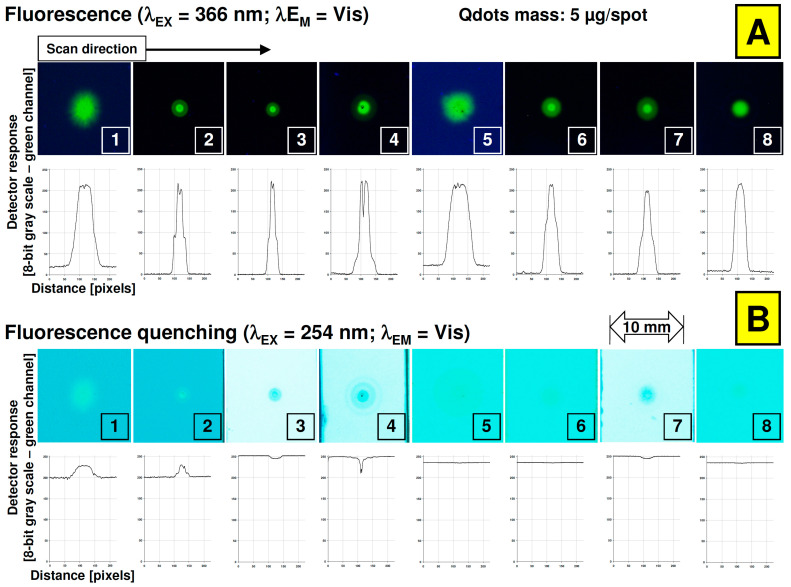
Signal response from QD spots registered on all stationary phases for given mass transferred to the plates—5 μg/spot and under different lighting conditions 366 nm (**A**) and 254 nm (**B**). Chromatographic stationary phases (plates) IDs: (1) Cellulose TLC, (2) RP-18 W HPTLC, (3) RP-18 W F_254_S HPTLC, (4) RP-18 F_254_ S HPTLC, (5) Polyamide 11 F_254_ TLC, (6) Silica gel 60 F_254_ TLC, (7) Silica gel 60 W F_254_ S TLC, (8) Aluminium oxide 60 F_254_ Type E TLC.

**Figure 4 nanomaterials-12-00745-f004:**
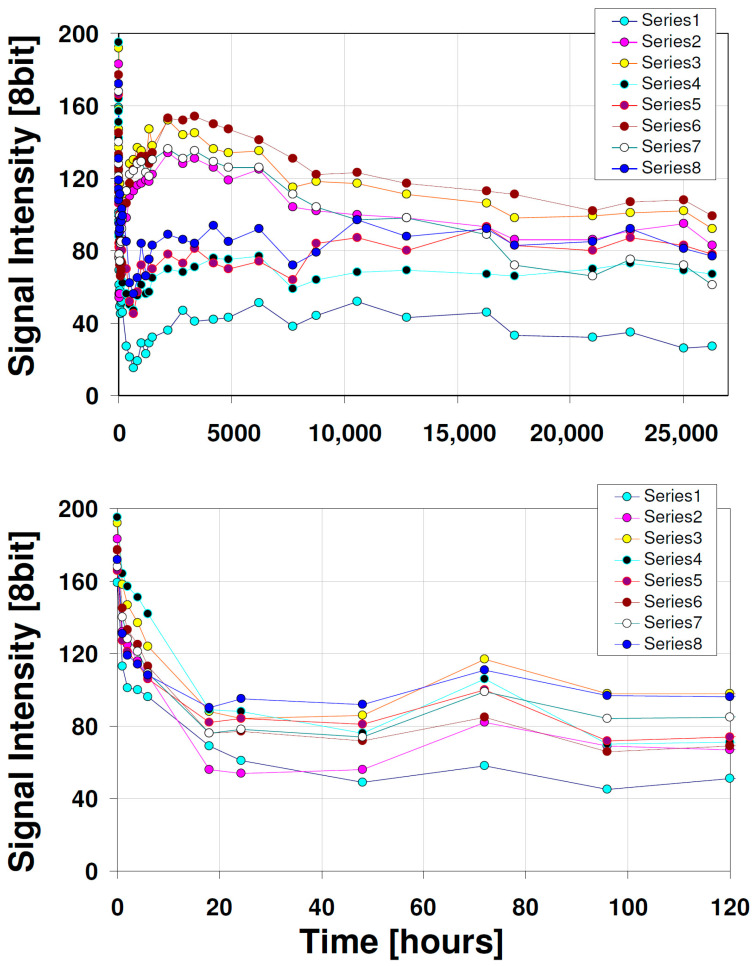
Fluorescence 366 nm signal intensity of QDs recorded over three years (**top**) and within first 120 h (**bottom**). Chromatographic stationary phases (plates) series IDs: (1) Cellulose TLC, (2) RP-18 W HPTLC, (3) RP-18 W F_254_S HPTLC, (4) RP-18 F_254_ S HPTLC, (5) Polyamide 11 F_254_ TLC, (6) Silica gel 60 F_254_ TLC, (7) Silica gel 60 W F_254_ S TLC, (8) Aluminium oxide 60 F_254_ Type E TLC.

**Figure 5 nanomaterials-12-00745-f005:**
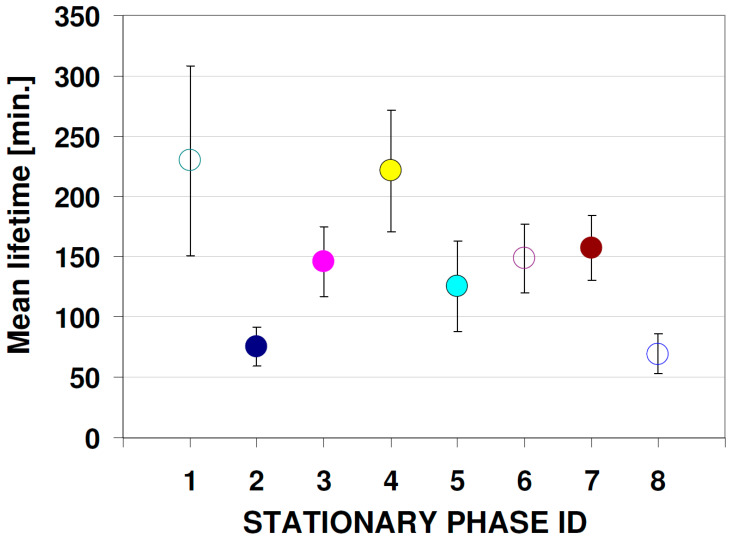
Mean lifetime values based on non-linear curve fitting (exponential decay with offset model: y = y_0_ + A_1_ e^−(x − x^_0_^)t^_1_) calculated for a 366-nm fluorescence signal within first 7200 min (120 h). Chromatographic stationary phases (plates) series IDs: (1) Cellulose TLC, (2) RP-18 W HPTLC, (3) RP-18 W F_254_S HPTLC, (4) RP-18 F_254_ S HPTLC, (5) Polyamide 11 F_254_ TLC, (6) Silica gel 60 F_254_ TLC, (7) Silica gel 60 W F_254_ S TLC, (8) Aluminium oxide 60 F_254_ Type E TLC.

**Figure 6 nanomaterials-12-00745-f006:**
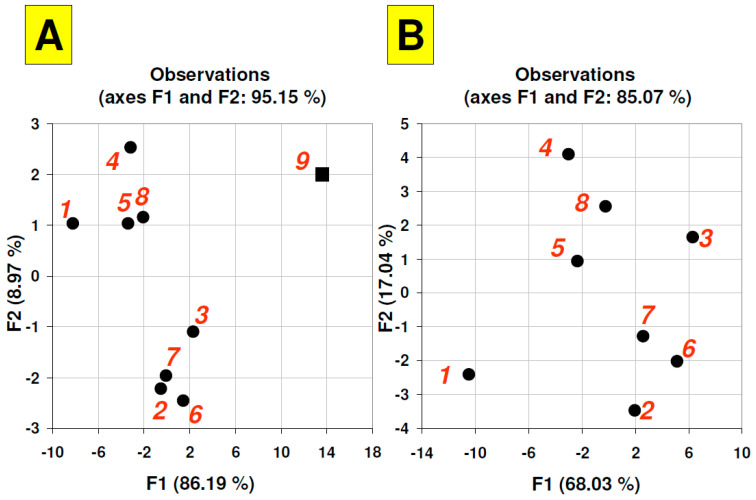
Principal component analysis—factor scores plots—related to fluorescence intensity (objects)/time (variables) matrix with (**A**) and without (**B**) fluorescence reference point (labelled as a black square). Chromatographic stationary phases (plates) series IDs: (1) Cellulose TLC, (2) RP-18 W HPTLC, (3) RP-18 W F_254_S HPTLC, (4) RP-18 F_254_ S HPTLC, (5) Polyamide 11 F_254_ TLC, (6) Silica gel 60 F_254_ TLC, (7) Silica gel 60 W F_254_ S TLC, (8),Aluminium oxide 60 F_254_ Type E TLC, reference area (9).

**Figure 7 nanomaterials-12-00745-f007:**
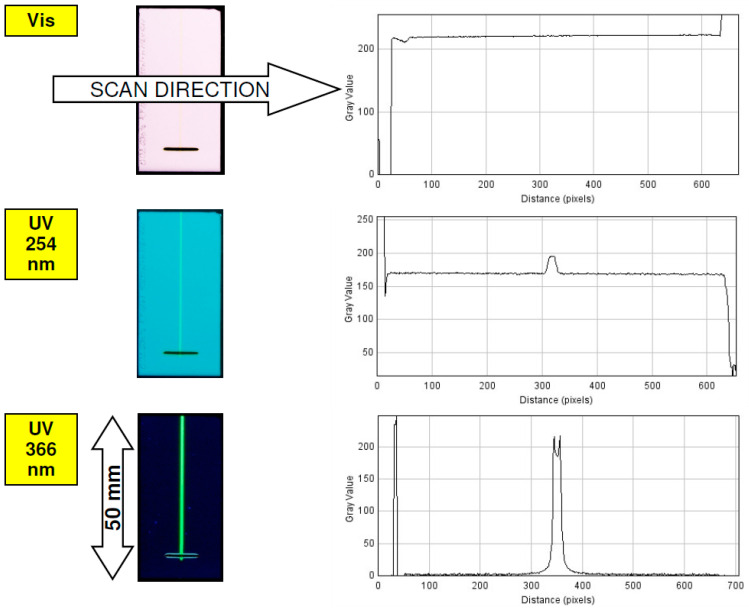
The signal intensity of QDs detection line formed on RP18W HPTLC plates before spirulina extract development. Densitograms presented on the right were generated horizontally to the mobile phase development.

**Figure 8 nanomaterials-12-00745-f008:**
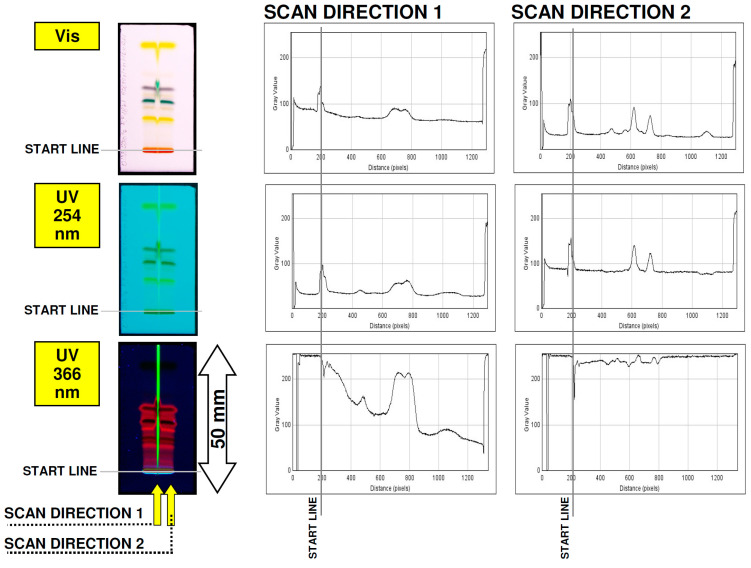
The signal intensity of the QDs detection line (scan direction 1) and raw extract (scan direction 2) formed on RP18W HPTLC plates after spirulina extract development. Densitograms presented on the right were generated vertically to the mobile phase development.

## Data Availability

The data presented in this study are available on request from the author.

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
