# Peer review of "Long-Term Fluorescence Behavior of CdSe/ZnS Quantum Dots on Various Planar Chromatographic Stationary Phases"

_nanomaterials, 2022, doi:10.3390/nano12050745_

Round 1

Reviewer 1 Report

The authors have estimated the stability of CdSe/ZnS quantum dots on the planar chromatographic stationary phases over 3 years. Eight planar chromatographic stationary phases including cellulose, octadecylsilane, polyamide, silica gel and aluminium oxide in different forms (TLC and HPTLC) were investigated. Time fluorescence behavior on different TLC/HPTLC supports was investigated using multivariate statistics, involving principal component analysis. In general, the experiment is well presented. It is recommended to be published after a major revision, and the following issues should be carefully addressed.  

  • The Abstract should be revised to be more concise, and the novelty is to be highlighted.
  • The fluorescence properties of CdSe/ZnS quantum dots should be estimated.
  • The authors claimed that plates polarity may influence the QDs fluorescence behavior on the planar stationary phases. A comprehensive discussion on this issue should be provided.

Author Response

The authors have estimated the stability of CdSe/ZnS quantum dots on the planar chromatographic stationary phases over 3 years. Eight planar chromatographic stationary phases including cellulose, octadecylsilane, polyamide, silica gel and aluminium oxide in different forms (TLC and HPTLC) were investigated. Time fluorescence behavior on different TLC/HPTLC supports was investigated using multivariate statistics, involving principal component analysis. In general, the experiment is well presented. It is recommended to be published after a major revision, and the following issues should be carefully addressed. 

Q1:     The Abstract should be revised to be more concise, and the novelty is to be highlighted.

Q2:     The fluorescence properties of CdSe/ZnS quantum dots should be estimated.

Q3:     The authors claimed that plates polarity may influence the QDs fluorescence behavior on the planar stationary phases. A comprehensive discussion on this issue should be provided.

REPLY TO THE COMMENTS PROVIDED IN REVIEW #1:

Q1. This part of manuscript has been extensively revised and improved.

Q2. Thank you for your comment regarding fluorescence properties. In this experiment the QDs fluorescence generated from planar chromatographic spots was acquired by CCD camera. Recorded signal was measured as a contrast value relative to background noise or dominating signal (depending on direct fluorescence or fluorescence quenching mode). Obtained values of such parameter in time domain were then considered for mathematical models or multivariate computations. In my future study I will focus more on given fluorescence properties and appropriately rearrange the experimental setup for such goal. In the present experimental setup an extensive discussion of additional fluorescence properties can be incremental and in fact not really change the finding presented, in my opinion.

Q3: Thank you for this question. Obviously, the effect of adsorbents polarity is just hypothesis or even sort of speculation based on the PCA pattern computed. This will be extensively tested in the future experiments concerning QDs behaviour on various complex solids, since analysis of more objects (solids) for multivariate analysis are necessary to link polarity (that is clearly different for different adsorbents) and QDs behaviour in time domain. In revised version a following sentence was modified: "This strongly suggest that support polarity may be a key factor for fluorescence behavior of QDs adsorbed, however, this hypothesis should be supported by additional experimental data."

Reviewer 2 Report

The manuscript described the long-time stability of CdSe/ZnS QDs on planar chromatographic stationary phases for micro-TLC analysis of Spirulina extraction. The data on QDs long-time stability were performed for 3 years, which were interesting for some readers, meanwhile the mentioned strategy which combined fluorescence quenching to micro-TLC analysis provide a new screen on the TLC technology. However, I think the manuscript should be carefully modified before publication according the following points:

  1. Whether the words “time domain” is proper in the title? I think the meaning in this work is the long-term stability of QDs attached on chromatographic stationary phases. I am not sure this can be involved in the concept of “time domain”.
  2. Page 8, Figure 1. In the caption, “General view of LumidotTM CdSe/ZnS quantum dots solutions (A; right column) and QDs spots on TLC plates (B; left column)” should be corrected to “General view of LumidotTM CdSe/ZnS quantum dots solutions (A; left column) and QDs spots on TLC plates (B; right column)”.
  3. Page 16, Figure 6, the dot lines for scan direction on the photos should be thinner to give more original picture information. I think the distance (x axis) in the right densitograms should correspond the direction from top to down in the left photos. It should be explained in the revision.
  4. Some recently published works on QDs are advised to refer in the revision, such as Angew. Chem. -Int. Ed. 2018, 57, 6216–6220; Colloids and Surfaces A 2021, 623, 126673; Colloids and Surfaces A 2021, 613, 126129.
  1. Language should be carefully modified especially in grammar. For example, (1) the abbr. μ-PADs in Page 2 and μPADs in Page 3 should be same one, and correspond same words in the Page 2 and Page 7. (2) A noun as an attributive is always in the singular, such as “quantum dots types” (Page 3), “For quantum dots stability study TLC plates array”, etc. (3) Page 8, “Proposed detection method involve 366 nm excitation”, here “involve” should be “involves”.

Author Response

The manuscript described the long-time stability of CdSe/ZnS QDs on planar chromatographic stationary phases for micro-TLC analysis of Spirulina extraction. The data on QDs long-time stability were performed for 3 years, which were interesting for some readers, meanwhile the mentioned strategy which combined fluorescence quenching to micro-TLC analysis provide a new screen on the TLC technology. However, I think the manuscript should be carefully modified before publication according the following points:

Q1:         Whether the words “time domain” is proper in the title? I think the meaning in this work is the long-term stability of QDs attached on chromatographic stationary phases. I am not sure this can be involved in the concept of “time domain”.

Q2:         Page 8, Figure 1. In the caption, “General view of LumidotTM CdSe/ZnS quantum dots solutions (A; right column) and QDs spots on TLC plates (B; left column)” should be corrected to “General view of LumidotTM CdSe/ZnS quantum dots solutions (A; left column) and QDs spots on TLC plates (B; right column)”.

Q3:         Page 16, Figure 6, the dot lines for scan direction on the photos should be thinner to give more original picture information. I think the distance (x axis) in the right densitograms should correspond the direction from top to down in the left photos. It should be explained in the revision.

Q4:         Some recently published works on QDs are advised to refer in the revision, such as Angew. Chem. -Int. Ed. 2018, 57, 6216–6220; Colloids and Surfaces A 2021, 623, 126673; Colloids and Surfaces A 2021, 613, 126129.

Q5:         Language should be carefully modified especially in grammar. For example, (1) the abbr. μ-PADs in Page 2 and μPADs in Page 3 should be same one, and correspond same words in the Page 2 and Page 7. (2) A noun as an attributive is always in the singular, such as “quantum dots types” (Page 3), “For quantum dots stability study TLC plates array”, etc. (3) Page 8, “Proposed detection method involve 366 nm excitation”, here “involve” should be “involves”.

REPLY TO THE COMMENTS PROVIDED IN REVIEW #2:

Q1. Thank you for this comment - the manuscript title has been corrected.

Q2. Figure 1 caption has been corrected.

Q3. Thank you for this comment. I understand that this question concerns the Figure 8 (not Figure 6). In the present version the dot lines were removed from the plate scans. To clarify scanning direction on densitograms the position of start line on both plates and densitograms were indicated. This should eliminate a potential confusion with the scanning direction along X axis.

Q4. References list has been updated.

Q5. Thank you for this comment. Manuscript language was carefully modified.

Round 2

Reviewer 1 Report

The manuscrpit is significantly improved after revision, and it is recommended to be accepted.